# Plasma-derived exosome-like vesicles are enriched in lyso-phospholipids and pass the blood-brain barrier

**Martin Jakubec**[1], **Jodi Maple-Grødem**[2,3], **Saleha Akbari**[2], **Susanne Nesse**[2], **Øyvind Halskau**[1], **Astrid Elisabeth Mork-Jansson**[2]*

**1** Department of Biological Sciences, Faculty of Mathematics and Natural Sciences, University of Bergen, Bergen, Norway, **2** Faculty of Science and Technology, Department of Chemistry, Biochemistry and Environmental Technology, University of Stavanger, Stavanger, Norway, **3** The Norwegian Centre for Movement Disorders, Stavanger University Hospital, Stavanger, Norway

* astrid.mork-jansson@uis.no

**Data Availability Statement:** All relevant data are within the manuscript and its Supporting Information files.

## Abstract

Exosomes are vesicles involved in intercellular communication. Their membrane structure and core content is largely dependent on the cell of origin. Exosomes have been investigated both for their biological roles and their possible use as disease biomarkers and drug carriers. These potential technological applications require the rigorous characterization of exosomal blood brain barrier permeability and a description of their lipid bilayer composition. To achieve these goals, we have established a 3D static blood brain barrier system based on existing systems for liposomes and a complementary LC-MS/MS and $^{31}$P nuclear magnetic resonance methodology for the analysis of purified human plasma-derived exosome-like vesicles. Results show that the isolated vesicles pass the blood brain barrier and are taken up in endothelial cells. The compositional analysis revealed that the isolated vesicles are enriched in lyso phospholipids and do not contain phosphatidylserine. These findings deviate significantly from the composition of exosomes originating from cell culture, and may reflect active removal by macrophages that respond to exposed phosphahtidylserine.

## Introduction

Delivery of therapeutic agents to the brain is challenging due to the blood-brain barrier (BBB), a highly selective membrane that separates circulating blood from the brain extracellular fluid in the central nervous system [1]. The stringent selectivity of the barrier makes treatment of neurological diseases notoriously difficult. In fact, more than 98% of small-molecule drugs and almost 100% of large-molecule drugs, including peptides, recombinant proteins, monoclonal antibodies, genes and short interfering RNAs (siRNAs) cannot cross the BBB [2]. In the last decade, exosomes have emerged as promising vehicles for the transport of therapeutics across the BBB, giving promise for treatment of Alzheimer disease, Parkinson disease, epilepsy, mental disorders and more [3,4].

Several mechanisms allow compounds to cross the BBB and also prevent harmful compounds from entering the brain, and their normal function is critical for proper neuronal

**Funding:** AM: The University of Stavanger Norway, https://www.uis.no/frontpage/#Research, The Norwegian Epilepsy Union Norway, https://www.epilepsi.no/ Valide TTO Stavanger Norway https://valide.no/en/home ØH: The Norwegian research counsil grant #240063 and #226244/F50 https://www.forskningsradet.no/en/ Bergen Research Foundation (BFS-NMR-1) https://mohnfoundation.no/?lang=en Sparebankstiftinga Sogn og Fjordane (509-42/16). https://www.sparebankstiftinga.no/ The funders had no role in study design, data collection and analysis, decision to publish, or preparation of the manuscript.

**Competing interests:** The authors have declared that no competing interests exist.

**Abbreviations:** EVs, extracellular vesicles; DLS, dynamic light scattering; SUC, Sequential Ultracentrifugation; SEC, size exclusion chromatography; SG, sucrose gradient; LC-MS, liquid chromatography-mass spectrometry; NMR, Nuclear Magnetic Resonance; PC, phosphatidylcholines; SM, sphingomyelin; PS, phosphatidylserine; PE, phosphoethanolamine; LPA, lysophosphatidic acid; LPC, lysophosphatidylcholine; BBB, blood brain barrier; CNS, central nervous system.

function [5]. Passive diffusion of water-soluble agents across the BBB is negligible due to the tight junctions between endothelial cells, while $O_2$ and $CO_2$ freely diffuse across the BBB. As bases are cationic, they interact with the negatively charged head groups of phospholipids, which gives them a higher affinity than acids to pass the BBB, resulting in more efficient diffusion of lipid-soluble molecules than compounds with polar surfaces [6]. High molecular weight compounds like peptides and proteins can cross BBB via receptor- or adsorptive mediated transcytotic mechanisms [6].

Exosomes are small vesicle-like entities ranging from 50–100 nm in diameter [7]. In contrast to micro-vesicles and apoptotic bodies that bud off the cell membrane of the parent cell, exosomes are released from the parent cell when multi-vesicular bodies fuse with the membranes of the parent cell [7,8]. Researchers have suggested many different functional roles of exosomes [9], ranging from cellular garbage vesicles [10] to immunological functions [11] and cell-to-cell communication [12–14]. In addition, the potential of exosomes as biomarkers of disease [15–18] and as therapeutic drug carries [19] has been investigated in recent years.

Exosomes show diverse mixtures of proteins, lipids and nucleic acids depending on their cell of origin [20]. Moreover, it seems that the exosomic population can be very diverse even when derived from one cell line. This variability is documented in established databases such as EVpedia, Vesiclepedia or Exocarta, which contain several thousand protein and mRNA entries for different populations of exosomes [19,21,22]. However, there is very little information about the lipid components of exosomes, and there is no information about the lipid composition of exosomes isolated from human plasma. A significant challenge in exosome research is that the method of isolation, the cell of origin, and method of downstream analysis will greatly affect the outcome of the analysis performed [20].

Here, we have isolated vesicles of exosomal size (EVs) originating from human plasma using qEV size exclusion columns, and verified their purity and expected characteristics using dynamic light scattering (DLS), UV-vis spectroscopy, electrophoresis and immunoblotting. We then analyzed their BBB permeability and endothelial cell uptake in a 3D culture system and demonstrate that the BBB permeability of the isolated EVs is comparable to that of liposomes. Finally, the lipid composition was quantitatively determined by combining LC-MS/MS and $^{31}$P nuclear magnetic resonance (NMR). Interestingly, it seems that EVs originating from human plasma completely lack PS and contain a higher abundance of lyso- phosphatidylcholine lipid species than EVs isolated from PC3 cells.

## Material and methods

### Preparation of blood plasma

Anonymized blood samples from healthy volunteers were obtained following standard procedures by qualified health professionals in accordance with ethics guidelines provided by the Regional Committees for Medical and Health Research Ethics (REK). Venous blood samples were collected from the cubital vein of fasting individuals into EDTA tubes via a BD-vacutainer using a butterfly extension. Plasma and serum were separated by centrifugation 2x 15 min at 2500 xg and 4°C, eliminating cellular components. The platelet-free plasma was aliquoted into cryo-vials and stored at -80°C.

### Isolation of EVs by size exclusion chromatography

EVs derived from human plasma were isolated by size exclusion chromatography (SEC) using qEV original size exclusion columns following the manufacturer's protocol and in accordance with earlier studies purifying exosome (iZON Science, Oxford OX4 4GA, United Kingdom, and [23]). In brief: Plasma samples were centrifuged 15 min at 2500 xg and 4°C and 13.000 xg

to remove cellular debris and proteins respectively. The qEV column was equilibrated with 10 mL degassed phosphate buffer saline (1x PBS) and let to stabilize to room temperature (RT). After loading the column with 0.5 mL plasma, fractions were eluted with degassed 1x PBS and collected.

## Analysis of EV size and homogeneity

DLS was applied to determine the size and homogeneity of three independent isolations of EVs using a Zetasizer Nano ZSP (Malvern, UK). In DLS analysis, the mean hydrodynamic diameter of particles ($h_d$) can be determined from the particles' characteristic Brownian motion. An intensity distribution of determined $h_d$-values documents the dispersity of the sample [24]. Such intensity distribution were recorded at 20˚C and plotted as a function of calculated $h_d$-values. The average value of the peak, weighted by the Y-axis parameter (mean hydrodynamic diameter) is given for all samples.

## Analysis of the EV protein-lipid abundance

To determine the total protein content versus vesicles in the isolated EV fractions, absorbance at 280 nm (protein) and 498 nm (vesicles) were determined in a Spextramax® Paradigm® microplate reader (Molecular Devices, CA, 95134, USA). Absorbance at 280 nm and 498 nm were plotted against the qEV-eluted fractions.

## Denaturing-PAGE and WB analysis of isolated EVs

Isolated EV fractions were separated on a 10% Bolt ready made gel (Thermofisher), stained by Coomassie Brilliant Blue (CBB) and transferred to nitrocellulose (NC) membranes for subsequent immunological identification of the exosome specific CD63, CD81 and Hsp70 proteins (monoclonal 1˚ anti-CD63, DC81 and Hsp70 produced in rabbit (1:1000) and 2˚ Ab Goat anti-Rabbit HRP (1:20000), SBI system biosciences, Catalog# EXOAB-KIT-1). Finally, the immuno-reactive proteins were identified in a ChemiDoc™ Toch Imaging System (Bio-rad).

## Staining of EVs

Isolated EVs were pelleted for 1 hour, 16 xg at 4˚C and the pellet resuspended and incubated in WGA488 stain (Biotium, 5 μg/mL, diluted in HBSS) for 30 minutes, dark at RT. After incubation the EVs were centrifuged for 30 min, 16 xg at 4˚C and stained EVs washed twice with PBS+. The pellet was diluted in 0.5 mL pBEC Assay buffer (HBSS, 25mM HEPES, 0.5% BSA). To confirm successful staining of EVs, the fluorescence was recorded in a SpectraMax at 490/515 nm.

## Isolation of primary porcine brain microvascular endothelial cells (pBECs)

pBECs were isolated from porcine brains from a local slaughterhouse (Nortura, Stavanger, Norway). Each brain was gently washed in ice-cold PBS prior to removal of the meninges. The grey matter was scraped off the brains using a sterile scalpel and transferred to a 500 mL flask containing isolation medium (DMEM/F12 (Gibco), 10% FBS (Sigma-Aldrich) and 1% Pen-strep (Life Technologies). The grey matter was homogenized using a loose pestle in a cell homogenizer and cells were thereafter disrupted using a tighter pestle prior to a 1/10 dilution in isolation media. The homogenate was filtered through a 100 μM filter (Corning Cell strainer, Nylon), and the filters incubated in digestion media (75% DMEM/F12 (Gibco), 18.000 U Trypsin (Sigma Aldrich), 42.000 U Collagenase type 2 (Life technologies) and 23.000 U DNase (Life technologies)) for 1 hour at 37˚C and 5% $CO_2$ on a shaker. The digested

homogenate was washed with 10 mL media, centrifuged 5 min and 250 xg at 4°C and the the pellet resuspended in 10 mL media. The washing step was repeated once. Following resuspension, the suspension was kept on ice for 5 min, allowing the debris to settle. The supernatant was carefully removed and the centrifugation step repeated. Cells were cryopreserved and stored in liquid nitrogen.

## BBB permeability assays

Primary rat astrocytes (gift, Prof. Dr. Morten Nielsen, University of Aarhus, Denmark) resuspended in astrocyte medium (DMEM Glutamax (Gibco), 10% FBS and 1% Pen-Strep) were seeded in 6 well plates pre-coated with poly-L-lysine (Boster). At the same time pBECS were sowed in transwells (TW, polycarbonate) pre-coated with collagen (Gibco) and fibronectin (Gibco). TWs were inserted in the 6 well astrocyte wells and incubated at 37°C and 5% $CO_2$ for 3 days or until the trans endothelial resistance (TEER) reached 400–600 Ω. For the permeability assay 0.5 mL stained exosomes were added to the apical side of the TW and 1.5 mL of assay buffer (HBSS, 25mM HEPES, 0.5% BSA) to the basal side. Plates were incubated for selected time points in the dark at 37°C, 100 rpm on an orbital shaker. After incubation, 400 µL samples were collected from the apical side and 1.4 mL from the basal side, and porcine Brain endothelial cells (pBECs) washed quickly with 2x assay buffer (4°C, HBSS, 25 mM HEPES pH 7.4 and 0.5% BSA)). 250 µL lysis buffer (1% triton X-100 with protease inhibitors) was added to the collected samples and left at RT for a minimum of 45 min and lysate was collected in a low-bind tube. Lysates (basal and apical side) were centrifuged at 14.000 xg, 10 min at 4°C.

For apical lysates, 200 µL supernatant was transferred to a 96-well plate, and the pellet resuspend in 400 µL assay buffer and 200 µL added to the 96-well plate. For the basal lysates, 200 µL supernatant was transferred to the 96-well plate and the pellet was resuspended in 280 µL assay buffer (5x concentrated) and 200 µL was added to the 96-well plate. Total fluorescence was set to 100% and the % retained in pBECs, the % passed pBECs and the % passed TW filter was calculated.

## Cell culture

A T75 flask was coated with collagen (10 ng/cm$^2$) and incubated at 37°C in a 5% $CO_2$ humidified incubator for 1 hour prior to use. Endo-GRO Basal medium was prepared according to the manufacturers instructions (Merck Millipore, SCME004) by addition of 10 mM L-Glutamine, 1 µg/mL Hydrocortisone Hemisuccinate, 0.75 U/mL Heparin Sulfate, 50 µg/mL Ascorbic acid, 5% FBS, 0.2% 5 ng/mL Endo-GRO-LS Supplement, 5 ng/mL Rh EGF and 1 ng/mL FGF-2). A vial of hCMEC/D3 cells (Merck) was thawed in a 37°C water bath, transferred to a sterile 15 mL conical tube followed by a drop-wise addition of Endo-GRO complete medium. Cells were centrifuged at 300 xg for 3 min at RT and the pellet was resuspended in 10 ml of pre-warmed hCMEC/D3 complete medium. Finally the cells were plated on the pre-coated T75 flask and incubated at 37°C in a 5% $CO_2$ humidified incubator until 80% confluence was reached.

## Uptake assays and confocal microscopy

hCMEC/D3 cells (Merck) were grown on pre-coated coverslips in a 12-well plate until 70% confluent. The medium was removed and the cells were stained by WGA640 (Biotum, 20ug/mL, diluted in HBSS) for 10 minutes and washed three times in HBSS. 0.5 mL pre-stained vesicles, or buffer alone, were added to each well and incubated with the cells for 20 minutes. The coverslip was washed twice with 4% paraformaldehyde (PFA) and left to dry in order to fix the

cells. 8 µl mounting medium (Vectashield, Antifade Mounting Medium with DAPI) was added to the microscope slide and the coverslip was placed cell side down on the slide. Mounted slides were stored in the dark at 4˚C until use.

Images were acquired using an inverted Nikon A1R confocal laser scanning microscope with a 60X Plan-Apo/1.20 NA oil objective. Excitations used laser lines at 408 nm, 488 nm, or 561 nm, and images were recorded at 425/475 nm, 500/550 nm, or 570/620 nm, respectively. Laser intensities/detection settings were kept constant between parallel images to enable sample comparisons. NIS-Elements imaging software 4.0 (Nikon, Japan) was used for image capture. Stacks of images were acquired with a 0.5 µm confocal slice. The slices view was used to display orthogonal XY, XZ and ZY projections of the image sequence.

## Mass spectrometry

EVs isolated, as described above, were freeze-dried, and the resulting powder films were resuspended in 100 µl of water:acetonitrile: isopropanol: dichloromethane mixture (1.5:2:3:3.5). Accurate mass LC-MS and MS/MS was done on a Thermo Q-Exactive mass spectrometer fitted with a Dionex Ultimate 3000 UPLC (Thermo Fisher, USA). Briefly, each sample was separated using an HSS C18 column (1.7 µm particle size, Waters) and reverse phase elution. Buffer A consisted of a 40:60 (v/v) ratio of acetonitrile: water and buffer B was a 10:90 (v/v) ratio of acetonitrile: isopropanol. Lipids were then separated using a multi-step gradient from 40% of solvent B to 100% of solvent B across 17 min. Sample injection was set to 10 µL and MS/MS analysis was done using an iterative exclusion method, with a total of 4 runs for each positive and negative mode [25]. The analysis was performed using a home-written MATLAB script, LipMat, available at GitHub: https://github.com/MarJakubec/LipMat, whose functionality and use is described in detail here [26]. Briefly, the LipMat script compared the detected m/z values and compared them to a mass library of lipids. The script then uses a scoring function to assign individual lipids. The function is based on the Greazy scoring function, described here [27]. LipMat script was tested on Avanti Lipid MAPS standards: 12:0–13:0 PC, 17:0–14:1 PC, 17:1 LPC, 12:0–13:0 PE, 17:0–14:1 PE, 17:0–20:4 PE, 12:0–13:0 PG, 12:0–13:0 PI, 17:0–14:1 PI, 17:0–20:4 PI, 12:0–13:0 PS, 17:0–14:1 PS, 17:0–20:4 PS, and cardiolipin mix 1. The fragmentation patterns of Avanti standards are available with LipMat script on GitHub: https://github.com/MarJakubec/LipMat. All spectrum assignments were manually confirmed.

## Nuclear magnetic resonance

EVs isolated as described above were freeze-dried, and the resulting powder films were dissolved in the Culeddu-Bosco "CUBO" NMR solvent system (a mixture of 1 ml dimethylformamide, 0.3 mL trimethylamine by volume, and 100 mg guanidinium chloride) [28]. $^{31}$P spectra were acquired on a Bruker BioSpin NEO600 spectrometer (instrument carrier frequency set to 242.93 MHz) equipped with cryogenic probe using inverse gated proton decoupling. Experiments were performed at 300K, consisted of 3072 scans per sample, and had an overall recovery delay of 8 s between scans to ensured full relaxation of the $^{31}$P nuclei. Processing was then performed using the Topspin 4.0.1 software package from Bruker Biospin. Before Fourier transformation, FIDs were apodized using an exponential line broadening window function of 3.0 Hz. Manual phase correction and automatic baseline correction were then applied. The chemical shift scale was calibrated by setting the most abundant phospholipid signal in the sample–phosphatidylcholine (PC)–to zero ppm, and all peaks were deconvoluted. Peaks were identified and assigned to individual phospholipids based on the chemical shifts provided in previously published work using the CUBO solvent [29]. A total of five independent exosome samples were collected.

## Results

### Characterization of vesicles derived from human plasma

SEC is a gentle exosome isolation method, suitable for downstream analysis, however the method risks co-isolating microvesicles and lipid species. Therefore, we analyzed the isolated EVs by SDS PAGE, CBB staining, western blot (WB), Abs spectrometry and DLS to determine the purity, size and homogeneity (Fig 1 and S1 Fig). The electrophoretic separation of plasma (P, Fig 1A) identified a protein band at approximately 60 kDa and a high molecular weight smear. MS analysis identified two different proteins in the 60 kDa band: Immunoglobulin IgM heavy chain and Fibrinogen β, both abundant plasma proteins (S1 Table). In comparison, isolated EVs (V, Fig 1A) did not contain protein concentrations detectable by CBB.

To investigate the presence of exosomes in the isolated EV fractions we performed WBs targeting the exosome specific proteins CD63, CD9 and Hsp70 (Fig 1B). For both CD63 and Hsp70, chemiluminescence was detected around 60 kDa (Fig 1B, lane 2, 3 and 5), and for CD9 two chemiluminescent bands were detected in the 40 kDa region (Fig 1B, lane 4). This indicates that there is a significant amount of exosomes in the isolated vesicles.

Next, we wanted to investigate the difference in the ratio of protein/lipids in plasma and isolated vesicles. The correlation of protein and lipids in plasma and isolated EVs were determined by absorbance spectroscopy at 280 nm (Fig 1C, grey bar) and 495 nm (Fig 1C, red bar) in a SpectraMax® Paradigm® Multi-Mode Micro-plate reader (Molecular devices). Following SEC the protein concentration in the EV fraction decreased by a factor of 9.75 (Fig 1C, grey bars). The large drop in A280 nm corroborated the non-detectable protein concentration

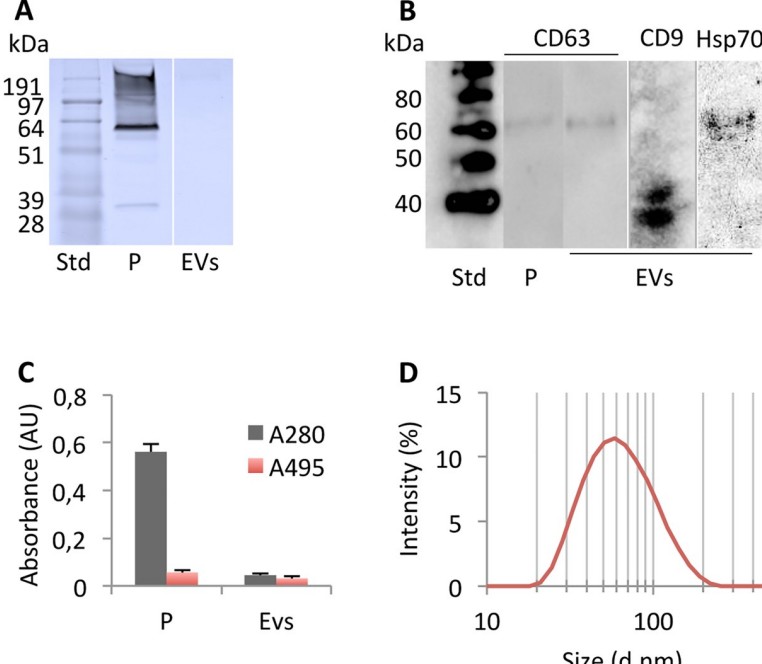

**Fig 1. Characterization of isolated EVs.** EVs originating from human plasma was isolated by SEC using qEV columns (iZON). Plasma (P) and EVs were separated by Bolt ready made 10% gels at denaturing conditions and stained with CBB (A). Proteins were transferred to a NC membrane and incubated in 1˚ Ab CD63, CD9 and Hsp70 produced in rabbit (1:1000) and 2˚ Ab Goat anti-Rabbit HRP (1:20000) (B). To determine the protein to vesicle ratio absorbance at 280 nm (grey) and 498 nm (red) were determined using a SpectraMax® Paradigm® Multi-Mode Microplate reader (Molecular devices) (C). The size and poly-dispersity of the isolated vesicles were analyzed in a Zetasizer Nano ZSP instrument (Malvern, UK) (D).

**Table 1. EVs BBB permeability.**

| Permeability | EVs (%) |
|---|---|
| Passed TW filter | 7.7 |
| Passed BBB | 1.3 |
| Retained apical side | 88.4 |
| Retained in pBECs | 2.6 |

in the CBB stained gel (Fig 1A, lane 2). The loss of EVs from the isolated vesicles compared to plasma was less pronounced, showing a decrease of only 1.63 fold (Fig 1C, red bars). The decrease in detected vesicles likely reflects elution of contaminating large apoptotic bodies and micro-vesicles.

Next, we investigated the size distribution and homogeneity of the isolated EVs to determine the main vesicle type present in the sample using DLS. Poly-dispersity indexes (PDI) between 0.1 and 0.3 are considered homogenous. Our analysis showed a homogenous sample with an average PDI of 0.21 ± 0.011 and with an average diameter of 68.8 ± 1.392 nm (Fig 1D). Apoptotic bodies, micro-vesicles and exosomes range from 1000–5000 nm.d, 100–1000 nm.d and 30–120 nm.d respectively. Thus, the average size and low PDIs of the isolated vesicles show that exosomes are abundant in the sample, but some co-isolation of micro-vesicles and small vesicles were observed, as sizes up to ≈ 164 nm.d was detected in the long tail of the data (Fig 1D). The plasma was to poly-dispersed to be measured by the instrument.

## BBB permeability and uptake by hCMEC/D3 cells

The EV BBB permeability was quantified in the static BBB model adapted to permeability assays of EVs (Table 1). Assays were set up using pBECs and rat astrocytes and showed that 7.8% of the vesicles passed the TW filter, indicating a high functionality of the adapted system (Table 1). Results showed a BBB permeability of 1.4% and retention of 88.4% for EVs, percentages comparable to that of liposomes [30,31]. Thus, the isolated EVs were able to pass the BBB in quantities acceptable for future use in transport of therapeutics across the BBB [30]. Interestingly, we also saw a high uptake (2.6%) of EVs in the pBECs, indicating a different delivery mechanism than seen for liposomes.

To further investigate the observed EV retention in the pBECs, uptake assays were performed in hCMEC/D3 cells and analysed by fluorescent confocal microscopy (Fig 2). EVs stained with WGA488 were visualised as clusters of different sizes (Fig 2, green). Subsequently, hCMEC/D3 cells were stained with the membrane dye WGA647 (red) and then incubated

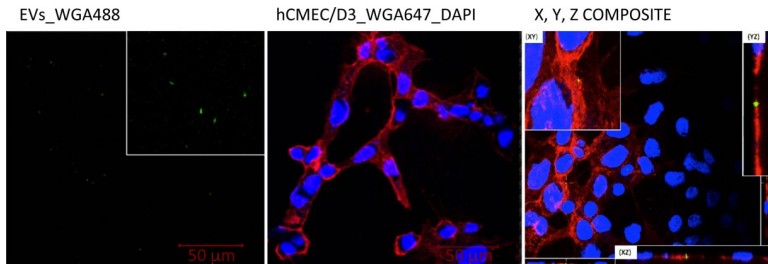

**Fig 2. EV uptake in endothelial cells (hCMC/D3) 15 min post addition of EVs the co-localization of EVs in the hCMEC/D3 cell line were visualized by confocal microscopy (Nikon A1).** EVs were stained with WGA 488 (green), nuclei with DAPI (Blue), cell membranes glyco-proteins with WGA640 (red). The images were analyzed by NIS element viewer version 4.40.00. Scale bare 50 μm.

with vesicles stained with WGA488 (Fig 2) or with buffer alone (Fig 2) for 15 minutes. The cells were then fixed and analyzed by confocal microscopy. The sample incubated with EVs revealed small green puncta to co-localise with the cells (Fig 2). Further analysis using serial sectioning revealed that the green vesicles were observed to be located in the same focal plane as the cells, suggesting import into the cells.

## Combined mass spectrometry and nuclear magnetic resonance phospholipid analysis

In order to analyze the phospholipid composition of EVs, we used a robust combination of NMR and LC-MS/MS (Figs 3 and 4). The [31]P NMR analysis provided reliable quantitative data on the phospholipids present in the samples, but no lipid acyl-chain information (Fig 3). The LC-MS/MS analysis provided a highly detailed picture of the lipids present in the sample, as well as quantitative data on the relative amounts of acyl-chains found on each lipid species (Fig 4).

The [31]P NMR revealed that the two most abundant lipid species present in exosomes lipid samples are phosphatidylcholine (PC) 76.7 ± 3.8% and sphingomyelin (SM) 20.3 ± 1.0% (Fig 3). Other phospholipid species, observed in some of the samples, includes phosphatidyletha-nolamine (PE), lyso-phosphatidylcholine (lyso-PC) and phosphatidylinositol (PI). However, those minor phospholipid species accounted for less than 3% of the total lipid content, and it was not possible to quantify their abundance using NMR.

The overall composition of the sample as determined by NMR was corroborated by LC-MS/MS (Fig 4 and S2 Table). We then proceeded with the analysis of fatty acid chains (FA) using LC-MS/MS, using the LipMat script whose application to cell lipidomics is described here [26]. Each MS2 spectrum that could be interpreted reliably according to the LipMat script and manual verification has been used to reconstruct the MS1 chromatogram for each lipid species. This reconstruction has then been used to quantify the relative abundance of each lipid for PC, SM, PI and PE (Fig 4). All identified lipid species are listed in S2 Table.

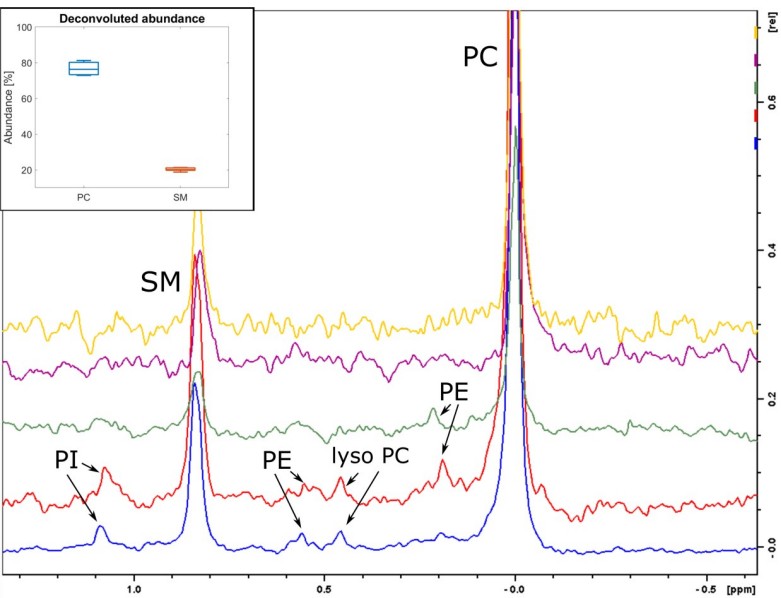

**Fig 3. [31]P NMR spectra of EV lipids.** Lipids isolated from EVs was analyzed for phospholipid content by [31]P NMR (Bruker NEO 600 MHz) and the most common phospholipids were identified. Figure insert: Box plot of the relative abundances determined by deconvolution of NMR spectra for the two most abundant phospholipids.

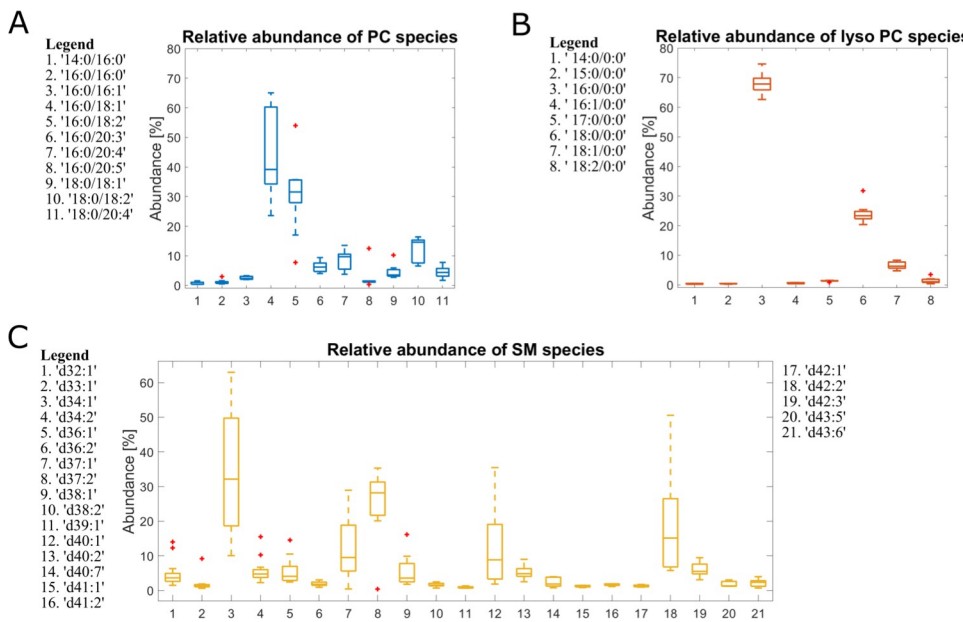

**Fig 4. MS analysis of lipid species abundance for PC, SM, PI and PE.** Lipids identified by MS2 has been quantified by MS1 chromatogram reconstruction. The bar plots represent the relative abundance of each lipid species towards other lipid species with the same headgroup. Centr3l mark indicates median, the bottom and top edges of the box indicates 25th and 75th percentiles, respectively. The whisker extends to the most extreme data points not considered outliers and the outliers are plotted as '+' symbol.

## Discussion

### Characterization of isolated vesicles

The characterization of EVs for future application in targeted delivery of therapeutics across the BBB is a research priority. Here we provide the first detailed characterization of EVs isolated from human plasma and show that they area able to cross the BBB.

To date, most of the exosomal membrane composition analysis has been performed on PC3 cells, B-lymphocytes and dendritic cells [32–34]. In one of the most detailed reports available, a quantitative lipidomic analysis of 70 nm exosomes originating from PC3 cells revealed a 1.36 ratio for lipids in the outer and inner membrane [33]. A typical feature of exosomes characterized so far is having relatively high abundances of cholesterol (28%), sphingomyelin (23%, SM) and phosphoethanolamine (13%, PE) when compared to the maternal cell [20].

While exosomes from PC3 cells are relatively well described concerning their lipid profile, very limited analyses have so far been performed on exosomes from biological fluids [20]. A literature review reveals two studies only: One focusing on prostasomes from seminal fluid isolated by sequential ultracentrifugation (SUC), size exclusion chromatography (SEC) and sucrose gradient (SG) followed by lipid analysis by LC-MS [35]. The second involves exosomes originating from urine, isolated by SUC, and analysed by MS [36]. However, both of these methods give fairly similar results with the three most abundant lipids being cholesterol, phosphatidyl serine (PS) and SM.

Contamination of abundant proteins, vesicles and lipids from plasma, serum or other biofluids is a major concern regarding purity and downstream analysis of EVs [37]. The gold standard for isolation of EVs has been ultracentrifugation, but this has recently been challenged by methods based on size, affinity and more [37]. In a recent study, Baranyai et al. made a qualitative and quantitative comparison of ultracentrifugation and size exclusion chromatography

(SEC) for isolation of blood-plasma derived exosomes [38], concluding on the advantage of SEC based exosome isolation from plasma samples. Further, SEC has resulted in significantly reduced amounts of contaminating albumin in the isolated EVs [39].

Here, CBB staining of plasma and isolated EVs resulted in non-detectable protein concentrations in the EV fraction (Fig 1A). Further, MS analysis could not identify albumin, immunoglobulin IgM heavy chain, fibrinogen β or any other abundant plasma protein in the isolated EV fraction (Table 1). The abundant plasma proteins could only be identified in the plasma fraction as expected. Further, we were able to demonstrate a 9.8 fold decrease in A280 nm between plasma and EVs showing efficient removal of plasma proteins, supporting the results from the CBB staining (Fig 1A and 1C). Our identification of the exosome-specific proteins CD63, CD9 and Hsp70 proteins (Fig 1B) supports a significant abundance of exosomes in the isolated vesicle fraction. Our DLS analysis determined average sizes for our vesicles to 58.3 d.nm and with a low PDI, suggesting that the isolated EVs primarily consists of exosomes. However, vesicles smaller than 30 nm.d and larger than 100 nm.d are present in the samples, likely reflecting a low degree of co-isolation of lipid structures and microvesicles (Fig 1D). The observed 1.6 fold decrease in vesicle content likely reflects removal of apoptotic bodies and large micro-vesicles, while the 8.9 fold drop in absorbance at 280 nm reflects the efficient removal of contaminating plasma proteins (Fig 1C). In accordance with the findings of Baranyai et al., our isolated EVs, resulted in homogenous vesicles with low protein contamination and limited co-isolation of other vesicles and lipids [38].

## EVs cross the BBB and are retained in endothelial cells

To date, most analysis of exosomes for therapeutic use in disease has been performed based on cell-culture medium [40]. To enable a more personalized strategy for treatment with EVs, EVs isolated from individual patients plasma/serum would be preferable to reduce immunological issues to a minimum. In addition, plasma/serum would be the most promising bio-fluid source due to the relatively high concentrations of EVs [41]. The BBB is a challenge when it comes to delivery of therapeutics to the central nervous system. We identified uptake of our plasma derived EVs in the hCMEC/D3 cells line (Figs 2 and 3), supporting the findings of Chen et. al 2016 who identified internalization of exosomes derived from HEK 293T cells in endothelial cells [40]. The same study show that the endothelial cells internalize the EVs by endocytosis via the trans-cellular route. The uptake mechanism for plasma-derived exosomes still remains to be solved. Our study show that a 15 min incubation of the EVs on the coverslip was sufficient for internalization of the EVs, while the exosomes originating from the HEK 293T medium where incubated for up to 48 hours to identify uptake in endothelial cells [40].

The 3D BBB permeability experiments were performed with primary porcine endothelial cells and rat astrocytes ensure physiological like tight junctions [42]. Our analysis showed a permeability of our plasma derived EVs equal to that observed for liposomes using absorbance spectrometry (Table 1). This is in partial contrast to what was observed for the HEK 293T cells [40]. Their study concludes that the HEK 293T cell culture derived exosomes can cross the BBB in an *in vitro* monolayer model only when the brain microvascular endothelial cells are induced by TNF-α and at much longer incubation times [40]. The differences between the analyses could be a result from the different isolation method, the incubation time, the model system and/or the EV origin.

## EVs originating from plasma do not contain phosphatidylserine

[31]P NMR analysis revealed that the main lipid components of the isolated EVs are PC and SM (Fig 3). This is the first time, to our knowledge, that EV species that do not contain

phosphatidylserine (PS) have been reported. Previously, it has been reported that PS is one of the three most common phospholipids found in exosomes [20] with concentrations varied between 15.6% (for HepG2/C3a, [43]) and 1.1% (for adipocytes, [44]). In our experiment, PS has been observed in neither [31]P NMR nor MS analysis, indicating that the concentration of PS in ventral blood EVs originating from plasma is negligible.

The lack of PS has functional implication in the role of human ventral blood EVs. PS is known to be located in the outer leaflet of activated blood cells, apoptotic bodies, micro-particles and micro-vesicles released from the plasma membrane [20]. PS presence activities monocytes that remove these particles from blood circulation [45]. Presence of PS on the surface of the EVs would cause them to be rapidly removed from circulation, which is not compatible with the role of exosomes in endocrine-like cell signaling [46]. The absence of PS then suggests that isolated EVs are not a waste byproduct, but an active component of blood plasma. It may also be that while PS may exist to some extent in freshly exported cell-derived EVs, these are removed by monocyte or macrophage activity from the blood and tissue, respectively.

## EVs contain a notably high abundance of lysophospholipids and monounsaturated fatty acyl chains

MS analysis (Fig 4) revealed the presence of lyso lipid species. Lyso-PC (16:0) was the third most abundant PC phospholipid. Other phospholipid species also showed a high abundance of lyso isoforms, with lyso-PI (18:0) and lyso-PE (16:0 and 18:0) being among the highest abundant lipid species within each head-group. Lyso-PI enrichment in exosomes has been reported previously [33]. However, there are no reports of the significant presence of lyso-PC and lyso-PE species in isolated exosomes.

Autotaxin is a secreted lipase that produces lysophosphatidic acid (LPA) and bind to exosomes [47]. Generation and delivery of LPA has been suggested to be mediated by exosomes and autotaxin [47]. The same study suggests that LPA is synthesised from lysophosphatidyl-choline (LPC) on exosome membranes. This finding supports our observed overrepresentation of LPC/LPA for the plasma derived exosome membranes analyzed in our study. In addition, our results support the notion that exosomes could function as a repository of lyso-species protecting them from lipases and thereby degradation [48].

For the two most abundant phospholipids, PC and SM, we have obtained detailed lipid species profiles (Fig 4). These revealed that the most abundant PC lipid is 16:0/18:1 and two out of three of the most abundant SM species are d18:1/16:0 and d18:1/19:0, i.e. lipid species containing one monounsaturated fatty acid chain. The high abundance of the lipid species containing one fully saturated and one monounsaturated fatty acid chains has been previously recognized as a key characteristic of exosomes from PC-3 cells [33] and Olie-neu cells [49]. These observations may also reflect the fact that 16:0 and 18:1 fatty acids are common substitutions on the lipids of eukaryotic cells. Lyso-species and monounsaturated phospholipids are essential in the management of curvature stress in the lipid bilayer [50,51]. It is possible that these components are vital in the correct formation and maintenance of small, high-curvature EVs. The management of curvature stress by lyso-PC species becomes more prominent in the perspective of missing PS, a structurally important, conically shaped lipid.

## Conclusion

We have successfully isolated and characterized EVs (predominantly exosomes) originating from human plasma. We concluded that the isolated EVs have a BBB permeability similar to liposomes but unlike liposomes they accumulate in endothelial cells. One of the most striking characteristics of the isolated exosomes is their lack of PS and high abundance of lyso species

for PC, PI and PE phospholipids. New insight into lipid composition of EVs/exosomes and the functional relevance of this is one of the necessary steps for improving our understanding of their biological and pharmacological properties.

## Supporting information

**S1 Fig. Raw images.**
(TIF)

**S1 Table. Details from MS analysis.**
(TIF)

**S2 Table. Details from lipid species analysis.**
(XLSX)

## Author Contributions

**Conceptualization:** Øyvind Halskau, Astrid Elisabeth Mork-Jansson.

**Data curation:** Martin Jakubec, Jodi Maple-Grødem, Saleha Akbari, Susanne Nesse, Øyvind Halskau, Astrid Elisabeth Mork-Jansson.

**Formal analysis:** Martin Jakubec, Jodi Maple-Grødem, Saleha Akbari, Susanne Nesse, Øyvind Halskau, Astrid Elisabeth Mork-Jansson.

**Funding acquisition:** Øyvind Halskau, Astrid Elisabeth Mork-Jansson.

**Investigation:** Martin Jakubec, Jodi Maple-Grødem, Saleha Akbari, Susanne Nesse, Øyvind Halskau, Astrid Elisabeth Mork-Jansson.

**Methodology:** Martin Jakubec, Jodi Maple-Grødem, Saleha Akbari, Susanne Nesse, Astrid Elisabeth Mork-Jansson.

**Project administration:** Astrid Elisabeth Mork-Jansson.

**Resources:** Martin Jakubec, Jodi Maple-Grødem, Øyvind Halskau, Astrid Elisabeth Mork-Jansson.

**Software:** Øyvind Halskau.

**Supervision:** Jodi Maple-Grødem, Øyvind Halskau, Astrid Elisabeth Mork-Jansson.

**Validation:** Martin Jakubec, Jodi Maple-Grødem, Saleha Akbari, Susanne Nesse, Øyvind Halskau, Astrid Elisabeth Mork-Jansson.

**Visualization:** Jodi Maple-Grødem, Susanne Nesse, Astrid Elisabeth Mork-Jansson.

**Writing – original draft:** Martin Jakubec, Jodi Maple-Grødem, Astrid Elisabeth Mork-Jansson.

**Writing – review & editing:** Martin Jakubec, Jodi Maple-Grødem, Saleha Akbari, Øyvind Halskau, Astrid Elisabeth Mork-Jansson.

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
