## [Decision Letter · Decision Letter 0]

31 Jul 2020

PONE-D-20-10605

Title - Plasma-derived exosome-like vesicles are enriched in lyso-phospholipids and pass the blood-brain barrier

PLOS ONE

Dear Dr. Mork-Jansson,

Thank you for submitting your manuscript to PLOS ONE. After careful consideration, we feel that it has merit but does not fully meet PLOS ONE’s publication criteria as it currently stands. Therefore, we invite you to submit a revised version of the manuscript that addresses the points raised during the review process.

The reviewers were very positive about the manuscript but have a few minor revisions suggested.

We look forward to receiving your revised manuscript.

Kind regards,

Joe W. Ramos, Ph.D.

Academic Editor

PLOS ONE

Journal Requirements:

2.PLOS ONE now requires that authors provide the original uncropped and unadjusted images underlying all blot or gel results reported in a submission’s figures or Supporting Information files. This policy and the journal’s other requirements for blot/gel reporting and figure preparation are described in detail at https://journals.plos.org/plosone/s/figures#loc-blot-and-gel-reporting-requirements and https://journals.plos.org/plosone/s/figures#loc-preparing-figures-from-image-files. When you submit your revised manuscript, please ensure that your figures adhere fully to these guidelines and provide the original underlying images for all blot or gel data reported in your submission. See the following link for instructions on providing the original image data: https://journals.plos.org/plosone/s/figures#loc-original-images-for-blots-and-gels.

3. Your ethics statement must appear in the Methods section of your manuscript. If your ethics statement is written in any section besides the Methods, please move it to the Methods section and delete it from any other section. Please also ensure that your ethics statement is included in your manuscript, as the ethics section of your online submission will not be published alongside your manuscript.

Reviewers' comments:

Reviewer's Responses to Questions

**Comments to the Author**

1. Is the manuscript technically sound, and do the data support the conclusions?

Reviewer #1: Yes

Reviewer #2: Yes

2. Has the statistical analysis been performed appropriately and rigorously? 

Reviewer #1: N/A

Reviewer #2: Yes

3. Have the authors made all data underlying the findings in their manuscript fully available?

Reviewer #1: Yes

Reviewer #2: Yes

4. Is the manuscript presented in an intelligible fashion and written in standard English?

Reviewer #1: Yes

Reviewer #2: Yes

5. Review Comments to the Author

Reviewer #1: The manuscript by Jakubec M et al., describes the composition and characteristics of extracellular vesicles isolated from the plasma of healthy individuals, and how this affects their ability to cross the blood-brain barrier (BBB). Despite the descriptive nature of this manuscript, there are several points of merit in this study. First, it represents the first extensive characterization of the properties of EVs isolated from the plasma of healthy individuals (which represents the necessary background knowledge to the use of EVs as therapeutics and/or biomarkers); second, it confirms (and quantifies) the ability of circulating EVs to cross the BBB; third, it shows for the first time that plasma EVs loose phosphatidylserine as a membrane component (a finding that is new and could have profound implications in specifically detecting their plasma origin); fourth, it provides a detailed and rigorous analysis of the lipid composition of EV membranes.

Only one minor comment: The Authors should provide the exact catalog number of the antibodies used for their EV characterizations.

Reviewer #2: Jakubee et. al., describe the isolation of exosomes from the plasma and its lipid components. The manuscript in general is well written and includes all the detail about the methods and approach. Reviewer has few minor comments:

1) The blots included in Figure 1B are not very clear and authors should improve the blots, probably better quality can be achieved by slot/dot blots.

2) Figure 1 is lacking statistical analysis for every panel. Authors should include this information.

3) For Figure 3, it is not clear the reviewer if any positive control was used to conclude that exosomes isolated from plasma do not containing PS.

4) For Figure 2, please include quantization and scale for images.

6. PLOS authors have the option to publish the peer review history of their article (what does this mean?). If published, this will include your full peer review and any attached files.

Reviewer #1: **Yes: **Muller Fabbri

Reviewer #2: No

---

## [Author Response · Author response to Decision Letter 0]

21 Aug 2020

Dear Joe W. Ramos, Ph.D

I here provide our response to the review of:

Journal Requirements:

Answer: We have changed the figure file titles, the font size of the major headlines and made sure the whole manuscript is written in Times New Roman. The manuscript now conforms to the journal style.

Answer: We have attached the original uncropped figure file for the gel and blot in Figs 1A and 1B, and we have made sure that the labeling in the figure adhere to the journal requirements. The file has been renamed and you will now find it attached as “S1_raw_images’.pdf” and sited in the supporting information (Pages 23 and 24, Revised Manuscript with Track Changes)

Answer: We have now stated in the Cover letter that the blot/gel image data can be found in the supporting information (Pages 23 and 24, Revised Manuscript with Track Changes).

3. Your ethics statement must appear in the Methods section of your manuscript. If your ethics statement is written in any section besides the Methods, please move it to the Methods section and delete it from any other section. Please also ensure that your ethics statement is included in your manuscript, as the ethics section of your online submission will not be published alongside your manuscript.

Answer: The ethics statement has been removed from the end of the manuscript (page 18, Revised Manuscript with Track Changes) and does now only appear in the Methods section (page 4, Revised Manuscript with Track Changes).

Answer: We have now inserted captions for our supporting information files at the end of the manuscript (pages 23 and 24, Revised Manuscript with Track Changes).

Reviewer #1

The manuscript by Jakubec M et al., describes the composition and characteristics of extracellular vesicles isolated from the plasma of healthy individuals, and how this affects their ability to cross the blood-brain barrier (BBB). Despite the descriptive nature of this manuscript, there are several points of merit in this study. First, it represents the first extensive characterization of the properties of EVs isolated from the plasma of healthy individuals (which represents the necessary background knowledge to the use of EVs as therapeutics and/or biomarkers); second, it confirms (and quantifies) the ability of circulating EVs to cross the BBB; third, it shows for the first time that plasma EVs loose phosphatidylserine as a membrane component (a finding that is new and could have profound implications in specifically detecting their plasma origin); fourth, it provides a detailed and rigorous analysis of the lipid composition of EV membranes.

Only one minor comment: The Authors should provide the exact catalog number of the antibodies used for their EV characterizations.

Answer: We agree with the reviewer and have now provided the exact catalog number of the antibodies in the Methods part of the manuscript (Page 5, in Revised Manuscript with Track Changes).

Reviewer #2: 

Jakubee et. al., describe the isolation of exosomes from the plasma and its lipid components. The manuscript in general is well written and includes all the detail about the methods and approach. Reviewer has few minor comments:

1) The blots included in Figure 1B are not very clear and authors should improve the blots, probably better quality can be achieved by slot/dot blots.

Answer: We agree with the reviewer that the blots should be improved. However the need to collect more plasma samples are very challenging due to the ongoing pandemic. The original blots provided in file S1_raw_images’.pdf, show that although there is background noise, it is still well possible to differentiate signal and noise. We therefore hope that the reviewer can accept the blots as they are.

2) Figure 1 is lacking statistical analysis for every panel. Authors should include this information.

Answer: We partially agree with the reviewers comment. In panel 1A, we have reported on the presence of protein in general by a coomassie brilliant blue stain. We do not use the data in panel 1A to support a quantitative statement and therefore we feel that statistical analysis is not needed. In panel 1B, we have presented the identification of the exosome specific proteins CD63, CD9 and Hsp70. We do not use the data in panel 1B to support any quantitative statement and therefore we feel that statistical analysis is not required. For panel 1C, we agree with the reviewer that some statistical analysis should have been included. We have now included error bars for three independent isolations of EVs. For the DLS analysis in panel 1D, we have presented the mean hydrodynamic diameter of three independent vesicle isolations with three technical replicates. This has now been described in the methods part (Page 5, in Revised Manuscript with Track Changes) for clarity. We have not presented error bars in the figure due to the consecutive nature of the data on the X-axis. We had however, presented the mean hydrodynamic diameter and polydispersity index with standard deviations in the text (Pages 11 and 12, Revised Manuscript with Track Changes). 

3) For Figure 3, it is not clear the reviewer if any positive control was used to conclude that exosomes isolated from plasma do not containing PS.

Answer: We agree with the reviewer that it was not clear if we had tested the method on controls containing PS. The LIPMAP scripts were tested on Avanti Lipid MAPS standards, including standards containing a range of PS species with different acyl-chain substitutions. The manuscript is now updated (Page 9, Revised Manuscript with Track Changes) to explicitly state which lipids were used for positive identification of PS species and validation of the mass spectrometry method. In addition to this, we also can detect PS using NMR, which provides a screen for this lipid species by independent means. Neither methods detected PS species in the present case, but are well able to do so when PS is present, see e.g. Jakubec et al. http://dx.doi.org/10.1021/acsomega.9b03463. 

4) For Figure 2, please include quantization and scale for images.

Answer: The scale bare was included in the figure legend, but we have now added the value to the line in the image as requested. We have not performed a quantization of the analysis in Fig 2, we have therefore made sure not to state any quantitative claims in the text relating to the data.

---

## [Decision Letter · Decision Letter 1]

7 Sep 2020

Plasma-derived exosome-like vesicles are enriched in lyso-phospholipids and pass the blood-brain barrier.

PONE-D-20-10605R1

Dear Dr. Mork-Jansson,

We’re pleased to inform you that your manuscript has been judged scientifically suitable for publication and will be formally accepted for publication once it meets all outstanding technical requirements.

Kind regards,

Joe W. Ramos, Ph.D.

Academic Editor

PLOS ONE

Additional Editor Comments (optional):

Reviewers' comments:

Reviewer's Responses to Questions

**Comments to the Author**

1. If the authors have adequately addressed your comments raised in a previous round of review and you feel that this manuscript is now acceptable for publication, you may indicate that here to bypass the “Comments to the Author” section, enter your conflict of interest statement in the “Confidential to Editor” section, and submit your "Accept" recommendation.

Reviewer #2: All comments have been addressed

2. Is the manuscript technically sound, and do the data support the conclusions?

Reviewer #2: Yes

3. Has the statistical analysis been performed appropriately and rigorously? 

Reviewer #2: N/A

4. Have the authors made all data underlying the findings in their manuscript fully available?

Reviewer #2: Yes

5. Is the manuscript presented in an intelligible fashion and written in standard English?

Reviewer #2: Yes

6. Review Comments to the Author

Reviewer #2: The authors have addressed all my comments and I have no additional comments. I recommend this manuscript for publication.

7. PLOS authors have the option to publish the peer review history of their article (what does this mean?). If published, this will include your full peer review and any attached files.

Reviewer #2: No

---

## [Editor Report · Acceptance letter]

11 Sep 2020

PONE-D-20-10605R1 

Plasma-derived exosome-like vesicles are enriched in lyso-phospholipids and pass the blood-brain barrier. 

Dear Dr. Mork-Jansson:

I'm pleased to inform you that your manuscript has been deemed suitable for publication in PLOS ONE. Congratulations! Your manuscript is now with our production department. 

Kind regards, 

on behalf of

Dr. Joe W. Ramos 

Academic Editor

PLOS ONE